# Farmers' Knowledge, Attitude, and Practices Regarding the Use of Agroecological-Based Pest Management Practices in Crucifers and Traditional African Vegetable (TAV) Production in Kenya and Tanzania

Essy C. Kirui [1,2], Michael M. Kidoido [1,*] , Daniel M. Mutyambai [1] , Dickson O. Okello [2] and Komivi S. Akutse [1]

[1] International Centre of Insect Physiology and Ecology, Nairobi P.O. Box 30772-00100, Kenya; ekirui@icipe.org (E.C.K.); dmutyambai@icipe.org (D.M.M.); kakutse@icipe.org (K.S.A.)

[2] Department of Agricultural Economics and Agribusiness Management, Egerton University, Njoro P.O. Box 536-20115, Kenya; dickson.okello@egerton.ac.ke

\* Correspondence: mkidoido@icipe.org

**Abstract:** Crucifers and traditional African vegetables (TAVs) are important to smallholders in Kenya and Tanzania, but yield remains below potential due to pests and diseases. Agroecological production methods present a nature-based solution to pest and disease management in crucifer and TAV production. We explore the status of farmers' knowledge, attitudes, and practices regarding agroecological-based production pest management practices. Structured and pretested questionnaires were used to collect data from 1071 vegetable farming households in Kenya and Tanzania. Using descriptive statistics, parametric, and non-parametric analysis, our study revealed that less than 20% of farmers had received training on agroecological-based practices and less than 25% were aware of most of these practices. Among those who were aware of the practices and could confirm their effectiveness less than 12% had adopted them, except for crop rotation and handpicking of pests. This study attributes the low adoption to farmers' negative attitudes towards the practices. Nonetheless, the study further revealed that training significantly and positively influences the adoption of the practices. Therefore, we recommend that governments and other stakeholders promote targeted awareness campaigns and increase access to training on vegetable production using sustainable pest and disease management practices.

**Keywords:** agroecological pest management; cruciferous vegetables; traditional African vegetables; sustainable agriculture; environmentally friendly agriculture systems; biodiversity conservation; Kenya; Tanzania

## 1. Introduction

Horticulture is the second-largest foreign exchange earner in Kenya, and the country is the largest supplier of vegetables to European Union (EU) markets [1]. Similarly, Tanzania's horticulture sector is growing at 11% per annum, surpassing its agricultural growth rate of 4%, and the country is one of the world's top 20 producers of vegetables [2]. Kenyan and Tanzanian households commonly cultivate cruciferous vegetables such as cabbages and kales [3–5]. Traditional African vegetables (TAVs) have also been part of the food systems of these countries for generations since they are exceptional sources of vitamins, dietary fiber, and minerals [3,6]. The most popular traditional African vegetables found in Kenya and Tanzania's urban and rural markets include amaranth (*Amaranthus* spp.), spider plant (*Cleome gynandra*), jute mallow (*Corchorus olitorius*), cowpea leaf (*Vigna unguiculata*), African nightshade (*Solanum scabrum.*), and African eggplant (*Solanum macrocarpon*) [7,8].

Crucifers and TAVs are essential in enhancing food security, improving nutrition, and mitigating health risks. These vegetables are nutrient-dense with great potential to reduce

malnutrition, the most important risk factor for disease [9–11]. Their production also creates sustainable streams of income for smallholder farmers who are their major producers [9,11]. However, in most sub-Saharan African countries, actual yields of vegetables are much lower than their potential [12]. This is because smallholder vegetable production faces a myriad of persistent challenges including pests, diseases, limited agricultural technology, lack of knowledge and skills on good agronomic practices, and adverse effects of climate change [12,13].

According to Srinivasan et al., smallholder farmers in sub-Saharan Africa commonly resort to blanket pesticide spraying in a bid to safeguard crops, particularly high-value horticultural crops. However, this practice frequently results in the misuse of pesticides, involving high doses and frequencies [14]. Globally, approximately 2.3 billion kilograms of pesticides are applied annually, but less than 1% reach the intended pests, while the majority find their way into the soil, air, and water systems [15]. This exerts negative impacts on humans, the environment, and on some beneficial organisms. In addition, the costs of these chemicals are also high and can burden smallholder farmers, resulting in reduced incomes [16]. Managing these production challenges faced by smallholder vegetable farmers consequently calls for innovative solutions that promote the use of ecologically and economically sustainable agricultural systems [17].

Agroecological approaches to farming are crucial for sustainable agricultural pests and disease management [18–20] and therefore have the potential to attenuate some of these challenges. Agroecological production refers to the design and management of cropping systems and farms based on socioeconomic, agronomic, and ecological principles, integrating indigenous, scientific, and experiential knowledge [21–23]. Agroecological production methods include practices such as intercropping, crop rotation, and biological pest control that rely on ecosystem services as an alternative to external inputs [19,24,25]. Recent studies have indicated that these production methods increase crop yields, improve biodiversity conservation, stabilize production through increased diversification, increase farms' climate change resilience, reduce the dependency of farmers on external inputs, and therefore improve food security and the livelihoods of smallholder farmers [26–29].

Several studies underscore a transition toward agroecological production as a path to achieving numerous Sustainable Development Goals [29,30]. However, many of these practices are not achieving their maximum potential impact due to low rates of adoption by smallholder farmers in developing countries like Kenya and Tanzania [31–34]. Increasing accessibility and adoption of agroecological production methods will require major adjustments to policies, institutions, development agendas, and research [35]. Understanding and support for these changes are usually backed by a synthesis of the state of vegetable farmers' knowledge and awareness about agroecological production approaches, attitudes, and sustainable practices and how they are all interrelated [36–39]. Considering the local knowledge of farmers is essential in integrating scientific knowledge into their day-to-day challenges, leading to more effective farm management practices [37,38]. Furthermore, studying farmers' attitudes and their adoption patterns provides valuable insight into their decision-making processes, shedding light on their readiness to transition to agroecological production methods [38]. Ultimately, information on farmers' knowledge, attitudes, and practices allows for development programs to be more effectively deployed and adapted to the needs and wants of the community [40].

While significant research has focused on promoting the application of agroecological practices [41–44], farmers' knowledge, attitudes, and use of agroecological practices in vegetable production remain underexplored [13,45]. This study, therefore, intended to fill this gap by characterizing the knowledge and utilization of agroecological-based practices. Specifically, we aimed to (1) assess farmers' access to training and knowledge in agroecological production in Kenya and Tanzania; (2) examine farmers' attitudes towards agroecological production in both countries; and (3) evaluate the status of use of agroecological practices by farmers in both countries. Based on our results, we highlight some interventions that are necessary to improve support for vegetable agroecological production.

## 2. Materials and Methods

### 2.1. Study Area

The study was conducted in Kiambu and Murang'a counties, located in Central Kenya and the Arusha and Kilimanjaro regions within Northern Tanzania (Figure 1). Kenya and Tanzania were chosen for this study due to the significance of vegetable production in these countries and their pivotal role in the global vegetable trade [1,2]. The specific study areas were selected because they constitute some of the major smallholder crucifer and traditional African vegetable growing zones. Kiambu County covers 2499 km$^2$, of which 1878.4 km$^2$ are arable land with 21,447 hectares under food crops [46]. Murang'a County, on the other hand, covers a total area of 2558.8 km$^2$, with an average farm size of 6.5 hectares under large-scale farming [47]. Tanzania's dry land covers an area of 886,040 km$^2$ with 45 million hectares of this land being suitable for agricultural production and approximately 10 million hectares being under cultivation [48]. Arusha region covers 37,576 km$^2$ and is made up of seven districts, while Kilimanjaro region, which also has a total of seven districts, covers 13,250 km$^2$ and has a variable climate regime.

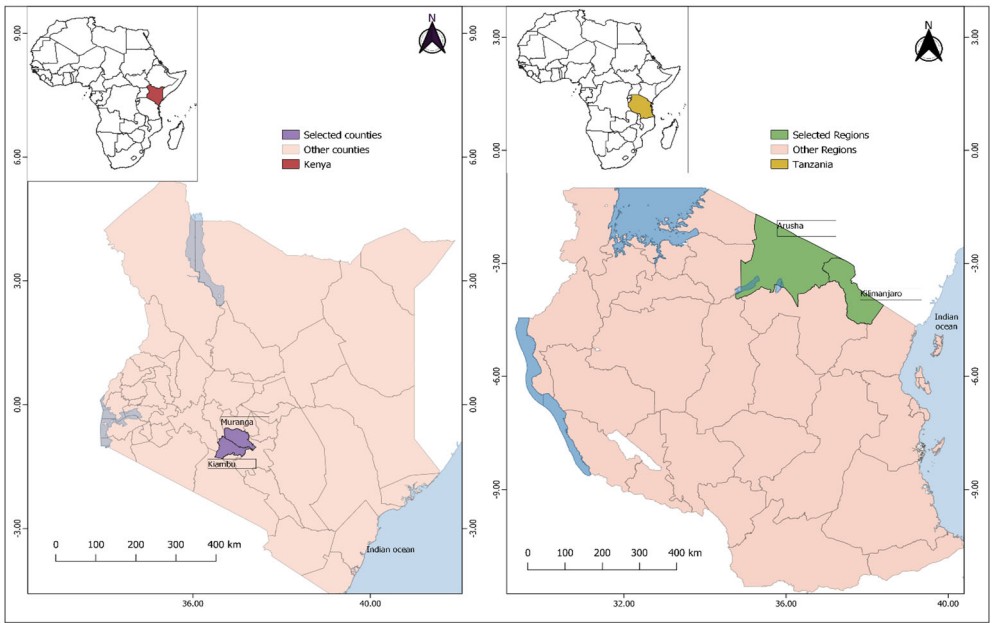

**Figure 1.** Map showing study areas in Kenya and Tanzania.

### 2.2. Sample Size

The sample size was determined using the [49] formula, specified as follows:

$$n_o = \frac{z^2 pq}{e^2} \tag{1}$$

where:

$n_o$ = sample size;

$z$ = standard value at a given confidence level ($\alpha$ = 0.05);

$p$ = the estimated portion of the population that has the attribute in question (maximum possible variance);

$q = 1 - p$;

$e$ = the desired level of precision.

The study requires a 95% confidence level and $\pm5\%$ precision. Since the population of all crucifer and traditional African vegetable farmers in the study areas is large and their

variability is unknown, a variability of *p* = 0.5 was assumed, which is approximately 50% of the total population. The resulting sample size was as demonstrated below:

$$n_o = \frac{(1.96)^2(0.5)(0.5)}{(0.05)^2} = 384 \text{ farmers} \qquad (2)$$

The derived sample was then estimated to be 384 farmers per country. However, to cater for errors in data collection or missing information from the target respondents, the sample size was increased from 384 to 550 in each country depending on the availability of respondents. During the actual survey, a total of 546 farmers were sampled in Kenya while 525 farmers were sampled in Tanzania. Therefore, the combined sample size for Kenya and Tanzania was 1071 farmers.

### 2.3. Sampling Method

The study employed a multistage sampling technique to select respondents in both Kenya and Tanzania. In the first stage, two counties in Kenya (Murang'a and Kiambu) and two regions in Tanzania (Arusha and Kilimanjaro) were purposively selected. Second, key informant interviews with county agricultural officers were used to purposively select five sub-counties of each selected county in Kenya, namely Kigumo, Kangema, Kandara, Murang'a South, and Kiharu in Murang'a; and Thika town, Gatundu South, Lari, Juja, and Githunguri in Kiambu. In Tanzania, key informant interviews with Intensified agroecological-based cropping systems to enhance food security, environmental safety, and income of smallholder producers of crucifers and traditional African vegetables in East Africa project (AGROVEG) staff and district agricultural officers were also used to purposively select eight wards in Arusha region, namely Akheri, Kikwe, Maroroni, Ngarenanyuki, Nkoanrua, Seela Sing'isi, Ambureni, and Knoanekoli and eight wards in Moshi District of Kilimanjaro region, namely North mwika, West Marangu, West Makuyuni, East Kahe, Kahe, Kirima, East Kibosho, and Njia Panda. In the third stage, villages were randomly selected within each sub-county and ward depending on the perceived population of crucifer and TAV farmers. Finally, 546 and 525 crucifers and TAV farming households in Kenya and Tanzania, respectively, were randomly chosen from all the selected villages.

### 2.4. Data Collection

Structured, pretested questionnaires that were programmed in CSPro were administered through computer-assisted personal interviews (CAPI) to solicit data from respondents. The questions were simple and gathered quantitative and qualitative information related to the knowledge, attitudes, and practices of crucifer and TAV farmers. The primary decision makers on vegetable production at the household level were targeted as the main respondents. The questionnaires were administered by a team of eight and nine enumerators in Kenya and Tanzania, respectively, who had vast experience in data collection and a good understanding of the local languages. These enumerators were also well-trained in data collection using CS entry. To assess farmers' knowledge, the questionnaire contained a series of closed-ended questions that prompted respondents to answer with a simple "Yes" or "No". These questions gauged their awareness of specific agroecological practices, the extent of their training, and the practical application of this training. To assess farmers' attitudes, the questionnaire employed closed-ended questions designed to uncover farmers' perceptions of agroecological practices and whether they implemented these practices in their cropping systems. Finally, close-ended questions that required farmers to identify different types of agroecological practices they have adopted were used to assess the current practices employed by smallholder farmers.

### 2.5. Data Analysis

The collected data were analyzed using the statistical software STATA 17, and the analysis entailed descriptive and inferential statistics. The descriptive statistics included

frequencies, percentages, cross-tabulations, and measures of central tendency. The study employed both parametric and non-parametric tests, which included the *t*-test, Pearson's Chi-square test, and Fisher's exact test. Since some of the data used in the *t*-test were skewed, the Wilcoxon rank sum test was carried out, and its findings/outputs corroborated those of the *t*-test. Since some of the expected cell values were less than five, Fisher's exact test was selected to investigate the potential correlation between agroecological practices used by trained and untrained farmers.

## 3. Results

### 3.1. Respondent Demographics

A total of 1071 crucifer and TAV farming households were selected for the study; 51% (546 households) were from Kenya and 49% (525 households) were from Tanzania (Table 1). Approximately 31% of the households interviewed were from Murang'a county, 20% were from Kiambu county, 27% were from Arusha region, and 22% were from Kilimanjaro region. Table 1 shows that most of the surveyed households were male-headed, 82% and 88% in Kenya and Tanzania, respectively. In Kenya, 38% of all the surveyed household heads (HHH) had completed primary education compared to 71% in Tanzania but a higher percentage, 41%, of the household heads in Kenya had completed secondary education compared to 14% in Tanzania. Most of the household heads interviewed, 47% in Kenya and 39% in Tanzania, stated that their main occupation activity was "other farming activity, apart from vegetable production" while only 8% in Kenya and 27% in Tanzania indicated that vegetable production was their main occupation. Chi-square analyses were carried out to test variations between the two countries regarding the gender, education, and occupation of the household head where all the tested variables were found to be statistically significant ($p < 0.01$) (Table 1). Figure S1 shows that the majority of crucifer farmers in Kenya and Tanzania grew kale and cabbages, respectively. On the other hand, TAV farmers in both countries mainly grew amaranth and African nightshade, respectively.

**Table 1.** Household demographic characteristics in Kenya and Tanzania.

| Variables | All | Kenya | Tanzania | $\chi^2$ Value | *p*-Value |
|---|---|---|---|---|---|
| Number of respondents by country | 1071 (100%) | 546 (51%) | 525 (49%) | - | - |
| Gender (HHH) | | | | | |
| Male | 910 (85%) | 448 (82%) | 462 (88%) | 8.823 | 0.003 *** |
| Female | 161 (15%) | 98 (18%) | 63 (12%) | | |
| Education level (HHH) | | | | | |
| None | 37 (3%) | 5 (1%) | 32 (6%) | | |
| Primary | 580 (53%) | 207 (38%) | 373 (71%) | | |
| Secondary | 298 (28%) | 224 (41%) | 74 (14%) | | |
| Tertiary | 119 (12%) | 109 (20%) | 10 (2%) | 93.467 | 0.000 *** |
| Secondary advanced | 6 (1%) | 1 (0%) | 5 (1%) | | |
| Vocational training | 31 (3%) | 0 (0%) | 31 (6%) | | |
| Occupation (HHH) | | | | | |
| Vegetable production | 186 (17%) | 44 (8%) | 142 (27%) | | |
| Other farming | 462 (43%) | 257 (47%) | 205 (39%) | | |
| Off-farm salaried | 135 (13%) | 93 (17%) | 42 (8%) | | |
| Casual labor | 58 (6%) | 27 (5%) | 31 (6%) | | |
| Own business off-farm | 149 (14%) | 76 (14%) | 73 (14%) | 35.036 | 0.000 *** |
| Retired | 54 (5%) | 38 (7%) | 16 (3%) | | |
| Unemployed | 11 (1%) | 0 (0%) | 11(2%) | | |
| Other | 16 (1%) | 11 (2%) | 5 (1%) | | |

Note: HHH = Household head; $\chi^2$ = Chi-square test; *** = significant at $p < 0.01$.

Sample *t*-tests were carried out to assess the equality in the means of some of the characteristics of respondents in the two countries, as presented in Table 2. The null hypothesis posited that the means of continuous variables of respondents from the two countries are the same. The findings from the study revealed significant differences between

Kenya and Tanzania in several key attributes (Table 2). In Kenya, the average age of the household head was 54 years, whereas in Tanzania it was 50 years. Furthermore, the average household size in Kenya was four people, compared to five people in Tanzania. In terms of gender distribution, Kenya had one male adult household member, while Tanzania had two male adult household members on average. The average total land cultivated in Kenya was 1.760 acres, while in Tanzania it was 1.340 acres. Regarding the quantity of insecticides and fungicides used per acre in one season, Kenya recorded an average of 450.472 milliliters of insecticides and 523.084 g of fungicides, whereas Tanzania reported higher quantities, with an average of 1086.462 milliliters of insecticides and 1179.726 g of fungicides. As indicated in Table 2, the differences between the two countries in household size, number of adult male household members, quantity of insecticides used, and quantity of fungicides used were negatively significant ($p < 0.01$). Conversely, the differences in the age of the household head and the total land cultivated were positively significant ($p < 0.01$). Additionally, the differences in the total number of adult female household members were negatively significant ($p < 0.05$).

**Table 2.** Results of independent sample *t*-tests of various continuous variables among surveyed households in Kenya and Tanzania.

| Variables | Kenya | | Tanzania | | *t*-Test |
|---|---|---|---|---|---|
| | **Mean** | **Std. Err** | **Mean** | **Std. Err** | |
| Age of household head | 53.630 | 0.540 | 50.040 | 0.530 | 4.730 *** |
| Household size | 4.216 | 0.080 | 4.821 | 0.090 | −5.110 *** |
| Number of adult female household members | 1.361 | 0.04 | 1.476 | 0.040 | −2.210 ** |
| Number of adult male household members | 1.337 | 0.040 | 1.550 | 0.040 | −3.770 *** |
| Total land cultivated (acres) | 1.760 | 0.060 | 1.340 | 0.070 | 4.550 *** |
| Quantity of insecticides used per acre in one season (milliliters) | 450.472 | 42.123 | 1086.462 | 109.6348 | −4.958 *** |
| Quantity of fungicides used per acre in one season (grams) | 523.084 | 67.116 | 1179.726 | 143.993 | −3.473 *** |
| Quantity of herbicides used per acre in one season (milliliters) | 944.790 | 267.574 | 865.113 | 161.358 | 0.267 |

Note: ** = significant at $p < 0.05$; *** = significant at $p < 0.01$.

### 3.2. Farmers Access to Training and Knowledge in Agroecological Production

To evaluate farmers' knowledge of agroecological production approaches, their awareness of various associated agroecological practices was assessed. The results show that a notably high percentage, 83% in Kenya and 65% in Tanzania, were aware of crop rotation (Figure 2). Around 65% and 36% of households in Kenya and Tanzania, respectively, were aware of handpicking infected/infested vegetables as an effective strategy for pest and disease control. Only 23% in Kenya and 7% in Tanzania knew that intercropping crucifers and TAVs with pest-repellant crops was a viable strategy for controlling pests. It was also noted that less than 10% of the farmers interviewed in both Kenya and Tanzania knew about the use of pest-resistant cultivars, pest predators, parasitoids, and biopesticides as pest control strategies in TAVs/crucifers. Only about 3% in Kenya and 13% in Tanzania knew about the use of insect traps as a pest control strategy in vegetable cropping systems.

The amount of training that farmers had received as well as the application of these trainings were also assessed to ascertain their knowledge on agroecological production. The results indicate that only 17% of the smallholder farmers had been trained on improved vegetable production in Kenya compared to 20% in Tanzania (Table 3). It was also found that approximately 16% of the farmers in Kenya had received training on integrated pest management (IPM) and conventional methods of vegetable pests' and diseases' management compared to 14% in Tanzania. Only 3% and 5% of these farmers in Kenya and Tanzania, respectively, stated that they had received training on intercropping as a method of controlling pests affecting crucifers and traditional African vegetables (Table 3).

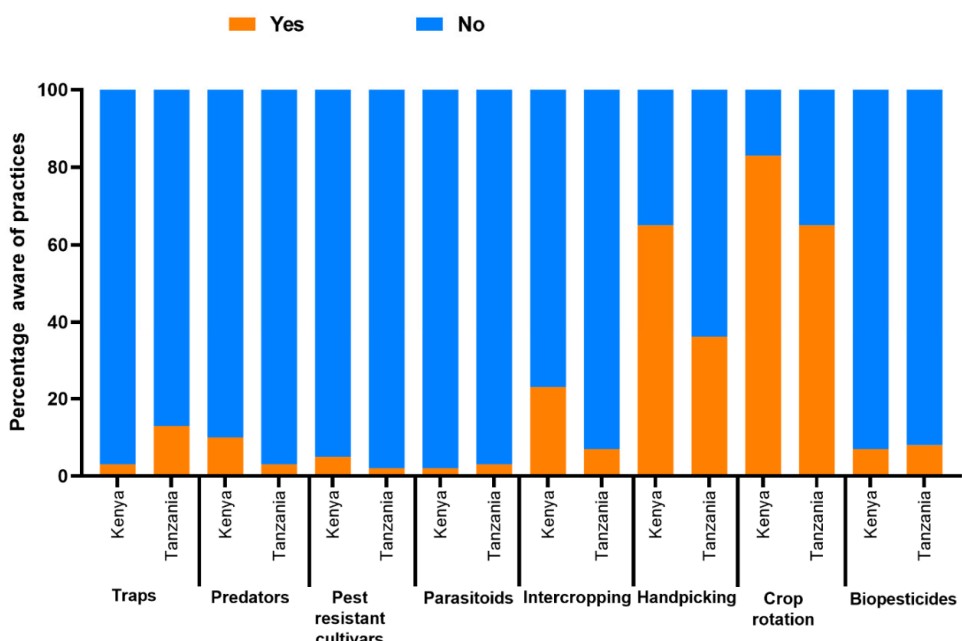

**Figure 2.** Percentage of farmers that were aware of agroecological practices in Kenya and Tanzania.

**Table 3.** Percentage of farmers that received, ever applied, and were currently applying training on improved vegetable production in Kenya and Tanzania.

| Training | % Received Training | | | % Ever Applied | | | % Currently Applying | | |
|---|---|---|---|---|---|---|---|---|---|
| | KE | TZ | All | KE | TZ | All | KE | TZ | All |
| Improved vegetable production methods | 17 | 20 | 18 | 15 | 14 | 12 | 12 | 12 | 12 |
| Vegetable pest and disease management (Conventional and IPM) | 16 | 14 | 15 | 12 | 10 | 10 | 10 | 9 | 10 |
| Intercropping as a vegetable pest control strategy | 3 | 5 | 4 | 1 | 2 | 1 | 0 | 1 | 1 |

Note: KE = Kenya; TZ = Tanzania.

The results in Table 3 also show that only 15% of all the farmers interviewed in Kenya and only 14% in Tanzania have ever applied improved vegetable production methods. Only 12% in Kenya and 10% in Tanzania have ever used vegetable pest and disease management strategies, while only 1% in Kenya and 2% in Tanzania have ever applied intercropping in controlling crucifer and TAV pests. Moreover, only 12% of all the farmers interviewed were found to be currently applying improved vegetable production methods in both Kenya and Tanzania. Only 10% of the farmers in Kenya and 9% in Tanzania were currently applying vegetable pest and disease management strategies, while only 1% of the farmers in Tanzania and even fewer in Kenya were currently applying intercropping in controlling pests and diseases.

Our results also reveal that most of this training had been conducted between the years 2017 and 2022 in both Kenya and Tanzania. In Tanzania, the number of trainings conducted was highest in the year 2020, while in Kenya, it was highest in 2022 (Figure S2). In addition, our findings show that the largest number of smallholder farmers in Kenya (15%) had received their training from the Ministry of Agriculture (MOA), whereas in Tanzania, 3% had received training from the Tanzania Agricultural Research Institute (TARI). A significant percentage (14%) of farmers in Tanzania had received their training from the World Vegetable Center. In both countries, fellow farmers were also found to be providing training to 5% of the farmers in Kenya and 6% in Tanzania. Around 24% of farmers in both Kenya and Tanzania received training from other organizations including Biovision and the Tanzania Horticulture Association (THA) (Figure S3).

### 3.3. Farmers' Attitudes and Practices in Agroecological Production

Farmers' perception of the effectiveness of agroecological practices was assessed to evaluate their attitudes towards these practices. The results show that more than 60% of all the farmers in both Kenya and Tanzania who had an awareness of the agroecological practices perceived them to be effective (Table 4). In addition, 98% perceived crop rotation and the use of biopesticides to be effective in Kenya and Tanzania, respectively. However, only 44% perceived the use of pest predators to be effective in Tanzania, while 69% perceived parasitoids to be effective in Kenya.

**Table 4.** Farmers' response to the effectiveness of agroecological practices in Kenya and Tanzania.

| Agroecological Practices | Kenya | | | Tanzania | | | Overall | | |
|---|---|---|---|---|---|---|---|---|---|
| | Obs. | No (%) | Yes (%) | Obs. | No (%) | Yes (%) | Obs. | No (%) | Yes (%) |
| Parasitoids | 13 | 31 | 69 | 17 | 41 | 59 | 30 | 37 | 63 |
| Predators | 54 | 13 | 87 | 18 | 56 | 44 | 72 | 24 | 76 |
| Bio-pesticides | 38 | 21 | 79 | 41 | 2 | 98 | 79 | 11 | 89 |
| Pest-resistant cultivars | 30 | 6 | 93 | 11 | 18 | 82 | 41 | 10 | 90 |
| Hand-picking infected vegetables | 354 | 8 | 92 | 187 | 43 | 57 | 541 | 20 | 80 |
| Crop rotation | 454 | 2 | 98 | 343 | 8 | 92 | 797 | 5 | 95 |
| Intercropping with pest-repellant crops | 126 | 12 | 88 | 39 | 28 | 72 | 165 | 16 | 84 |
| Traps | 19 | 5 | 95 | 70 | 41 | 59 | 89 | 34 | 66 |

Note: Obs. = Total number of farmers that were aware of each practice.

Our findings also indicate that about 76% and 61% of the farmers in Kenya and Tanzania, respectively, practice crop rotation as a strategy to control pests affecting their crucifers and TAVs (Figure 3). About 59% and 25% of the farmers in Kenya and Tanzania, respectively, practice hand-picking infected vegetables as a pest control strategy. Only 11% of the farmers in Kenya and 2% in Tanzania use intercropping with pest-repellant crops as a pest control strategy, while less than 5% in both countries use parasitoids, traps, predators, pest-resistant cultivars, and bio-pesticides for pest control (Figure 3).

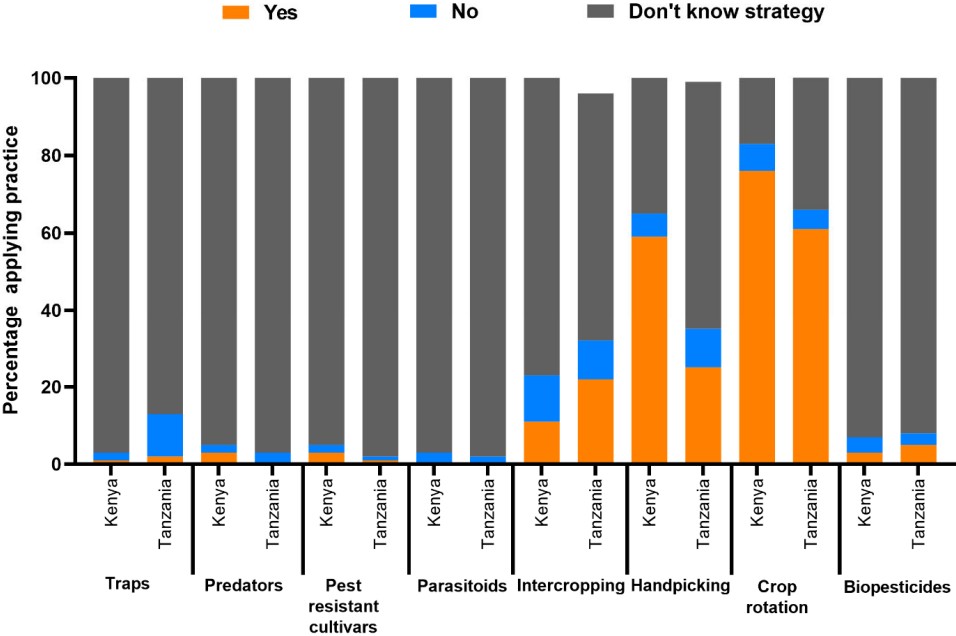

**Figure 3.** Percentage of farmers practicing each agroecological/alternative strategy in Kenya and Tanzania.

Regarding intercropping with pest-repellant crops, most of the farmers stated that they used marigolds, Ethiopian mustard, and onions to repel stem weevils and flea beetles, as shown in Table S1. Marigolds were also used to control stink bugs and amaranth stem weevils, while stem weevils were also known to be repelled by coriander. Only 27 farmers (0.03%) used pest traps out of all surveyed (Table S2). One farmer stated that he/she used marigolds to attract stem weevils while three farmers stated that they used onions to attract stem weevils as trap crops. About four farmers stated that they used onions to attract flea beetles and three other farmers used onions to attract amaranth stem weevils as trap crops for both pest species.

### 3.4. Comparing the Adoption of Agroecological Practices among Trained and Untrained Farmers

To determine whether there is any statistically significant relationship between agroecological practices adopted and farmers that received training, Fisher's exact tests were conducted for both Kenya and Tanzania. Results showed a statistically significant relationship ($p < 0.01$) between Kenyan farmers who received training on the use of predators, biopesticides, pest-resistant cultivars, crop rotation, intercropping with pest-repellant crops, and traps (Table 5). In addition, our findings also show that the relationship between Tanzania's farmers who received training on the use of handpicking infected/infested vegetables, crop rotation, and intercropping with pest-repellant crops was statistically significant ($p < 0.01$; Table 6). Table 6 also indicates that there was a statistically significant relationship ($p < 0.05$) for farmers who received training on the use of pest-resistant cultivars.

**Table 5.** Association between farmers' training and use of agroecological practices in Kenya.

| Training | Parasitoids | | Predators | | Biopesticides | | Agroecological Practices Pest-Resistant Cultivars | | Hand-Picking Infected Vegetables | | Crop Rotation | | Intercropping with Pest-Repellant Crops | | Traps | |
|---|---|---|---|---|---|---|---|---|---|---|---|---|---|---|---|---|
| | Not Adopt | Adopt | Not Adopt | Adopt | Not Adopt | Adopt | Not Adopt | Adopt | Not Adopt | Adopt | Not Adopt | Adopt | Not Adopt | Adopt | Not Adopt | Adopt |
| Not trained | 413 | 2 | 403 | 12 | 410 | 5 | 407 | 8 | 177 | 238 | 118 | 297 | 385 | 30 | 414 | 1 |
| Trained | 131 | 0 | 119 | 12 | 120 | 11 | 122 | 9 | 49 | 82 | 12 | 119 | 99 | 32 | 126 | 5 |
| Fisher's exact test (p-value) | 1.000 | | 0.005 *** | | 0.000 *** | | 0.008 *** | | 0.310 | | 0.000 *** | | 0.000 *** | | 0.004 *** | |

Note: *** = significant at $p < 0.01$.

**Table 6.** Association between farmers' training and use of agroecological practices in Tanzania.

| Training | Parasitoids | | Predators | | Biopesticides | | Agroecological Practices Pest-Resistant Cultivars | | Hand-Picking Infected Vegetables | | Crop Rotation | | Intercropping with Pest-Repellant Crops | | Traps | |
|---|---|---|---|---|---|---|---|---|---|---|---|---|---|---|---|---|
| | Not Adopt | Adopt | Not Adopt | Adopt | Not Adopt | Adopt | Not Adopt | Adopt | Not Adopt | Adopt | Not Adopt | Adopt | Not Adopt | Adopt | Not Adopt | Adopt |
| Not trained | 397 | - | 397 | 0 | 377 | 20 | 396 | 1 | 320 | 77 | 179 | 218 | 393 | 4 | 391 | 6 |
| Trained | 128 | - | 127 | 1 | 122 | 6 | 125 | 3 | 73 | 55 | 27 | 101 | 119 | 9 | 124 | 4 |
| Fisher's exact test (p-value) | - | | 0.244 | | 1.000 | | 0.047 ** | | 0.000 *** | | 0.000 *** | | 0.001 *** | | 0.268 | |

Note: ** = significant at $p < 0.05$; *** = significant at $p < 0.01$.

## 4. Discussion

The purpose of this study was to explore and understand the knowledge, attitudes, and practices of farmers in vegetable agroecological production systems. An assessment of farmers' demographics was conducted to provide an understanding of the social and economic context that may influence farmers' knowledge, attitudes, and practices. The results indicate significant heterogeneity in terms of gender, education levels, and occupations observed between the surveyed households in Kenya and Tanzania. These results affirm the necessity of research into the knowledge, attitudes, and practices of smallholder farmers that consider their unique characteristics and their local reality [36]. The disparities uncovered through this analysis emphasize the need for targeted interventions and tailored approaches in promoting agroecological practices among farmers [38].

The findings reveal that most of the households in Kenya and Tanzania were male-headed, which is comparable to other studies [50,51]. The number of male-headed household in Kenya was also found to be significantly higher than in Tanzania. This observation connotes important gender imbalances that impact the adoption of agroecological practices

and therefore need to be responded to through awareness creation. Previous studies have revealed that male-headed households tend to be faster at adopting certain agroecological strategies [52], while female households lagged in the adoption of modern practices [53]. Reynolds et al., however, found that female-headed households were more likely to plant diverse crops per hectare [50]. There is, therefore, a need for gender-sensitive policies and interventions that could promote equal access to productive resources in the agriculture sector, particularly in TAV and crucifer production systems [54]. Additionally, most of the household heads in both countries were found to have attained at least a primary-school level of formal education, although education levels in Kenya were significantly higher than in Tanzania. These education levels indicate that most of the farmers can read and write and may therefore accept new ideas and knowledge more avidly [55]. However, most of these household heads were also found to be carrying out other farming activities besides vegetable production as their main occupation, which could hinder their adoption of some agroecological practices that require a lot of time in their implementation [56].

The study revealed significant variations between Kenyan and Tanzanian households in terms of the age of household head, household size, the number of adult male and female household members, as well as the quantity of insecticides and fungicides used per acre in one season. These significant differences in attributes provide valuable contextual information that is crucial for understanding farmers' knowledge, attitudes, and practices related to agroecological practices. Therefore, it suggests that any strategies aimed at promoting agroecological practices need to be customized to address the specific needs and challenges faced by farmers in each country [38]. The average household size in Tanzania was found to be significantly larger than in Kenya with more adult male members, indicating higher labor endowment [56]. It was also revealed that although the average total land cultivated in Kenya was bigger than in Tanzania, the amounts of insecticides and fungicides used per acre in one planting season were significantly higher in Tanzania than in Kenya. Kapeleka et al. similarly documented that the use of pesticides in Tanzania is high and that it is escalating (58.4%) along with changing pesticide formulations [57].

Following Bloom's taxonomy as cited by Kusumawardani et al., knowledge is the first element of perception that generates attitudes and influences behavior [58]. The initial step towards acquiring agricultural knowledge involves having agricultural awareness, which refers to comprehension of basic agricultural concepts [59]. Our findings reveal that most of the farmers were aware of cultural control practices such as crop rotation and handpicking infected vegetables. This supports the findings of Chepchirchir et al. [60], who conducted a study in Kenya and Uganda and discovered that farmers were cognizant of these practices. These findings highlight the preservation of some indigenous knowledge on pest management within the communities surveyed. Our results, however, show that only a few farmers were aware that intercropping crucifers and traditional African vegetables with pest-repellant cultivars could present viable pest- and disease-control strategies. These findings concur with Laizer et al., who reported that only a handful of all the farmers surveyed in Northern Tanzania were aware that intercropping can be used as a strategy for controlling pests and weeds of common beans [61]. A study by Machekano et al. in Botswana also found that 70.6% of the farmers were not aware of suitable intercrops for cruciferous vegetables that could reduce pest infestations by repellence, interference, or otherwise [62]. These results underscore the need for targeted initiatives to educate farmers on intercropping techniques.

The results of our study further reveal that fewer farmers were aware of the use of pest-resistant cultivars. These results, however, slightly differ from the findings of Chepchirchir et al., who reported that a relatively higher number (33%) of tomato small-holder farmers in the Kirinyaga and Kajiado counties of Kenya were aware of pest-tolerant varieties [60]. Additionally, our findings differ from those of Nampeera et al., who carried out an investigation of farmers' knowledge and practices in the management of leafy amaranth insect pests in Kenya and reported that none of the farmers interviewed mentioned

knowing the use of pest-resistant cultivars [63]. It was also discovered that only a few farmers were aware of biopesticides, which is consistent with the study by Nyangau et al. [64].

Our results also indicate that farmers are typically unaware of insect traps. These results, however, differ from those of Chepchirchir et al., who reported that farmers in both Kenya and Uganda were moderately aware of pheromones as well as sticky traps [60]. Our findings also reveal that very few of the farmers surveyed were aware of the use of pest predators and parasitoids as agroecological-based approaches for vegetable pest control. This is consistent with the results obtained by Mkenda et al., who observed a similar lack of awareness among Tanzanian farmers regarding natural enemies of insect pests. However, their study revealed a positive transformation in awareness following a training course and up to 80% of surveyed farmers recognized beneficial insects and expressed an intention to adopt this pest management technique [65]. Overall, these findings emphasize a need to increase education, training, and outreach efforts to enhance farmers' awareness and understanding of the various biological and agroecological-based cropping systems for controlling vegetable pests.

Training is particularly important in equipping farmers with awareness and knowledge in agroecological production, since this requires an attuned understanding of existing climatic and plant relationships [66,67]. Kanjanja et al. found that the awareness that farmers had of agroecological production increased after the farmers received training from extension officers and that this training had the potential to change their behavior and attitudes toward agroecological practices [33]. Our study revealed that despite commendable efforts by the government, non-governmental organizations, and farmers themselves to provide training in recent years, only a small number of the farmers in Kenya and Tanzania had participated in these programs. As a result, farmers' knowledge of agroecological production may be negatively impacted if training and awareness programs are not strengthened. However, many of these farmers had also stated that they were aware of practices such as crop rotation and handpicking of infected/infested vegetables. This implies that farmers also rely on other sources of information such as their observations, instincts, and experiences as sources of knowledge. Nonetheless, relying too heavily on these sources can result in the right information being missed [68].

When asked whether they had ever applied or are currently applying the training they received on improved vegetable production methods or pest- and disease-management techniques, very few farmers answered positively. Paradoxically, a significant majority reported carrying out crop rotation and handpicking infected/infested vegetables, both of which are practices related to vegetable production and pest and disease management. This suggests that most of the farmers were not aware that some cultural techniques could be useful in improving vegetable production and in managing pests and diseases. Similar results were also reported by Laizer et al., who carried out a study in Northern Tanzania on smallholder common bean farmers and discovered that they were implementing cultural control methods such as crop rotation and intercropping but the majority did not acknowledge them as methods for managing pests [61]. This further underscores the need to train farmers in the most effective pest management techniques for their specific crops and growing conditions.

While some farmers in both countries had received training on the intercropping of vegetables to control pests, only a few of them stated that they had ever applied this training on their vegetable farms, and a smaller proportion reported implementing it within the last 12 months before the survey. Farmers who employed intercropping farming frequently utilized certain crops to repel pests, but these same crops were also found to attract pests. Similar findings were reported by Masqood et al., who found that the most effective method for controlling flea beetles on Brinjal (eggplant) involves the use of coriander as an attractant intercrop and onion as a repellant buffer crop [69]. Similarly, Iamba and Yaubi also used marigolds to attract cabbage flea beetles [70]. However, Iamba [71] found that marigolds attract natural enemies against cabbage flea beetles, while onions did not have a significant effect on the abundance of the pest. There is therefore a need to carry out further

experiments in this area not only to elucidate these interactions but also to educate farmers on the appropriate repellant and attractant plants.

Our study revealed that almost all farmers did not practice the use of biological pest control methods, namely parasitoids, pest predators, biopesticides, pest-resistant cultivars, and traps, on their crucifer and TAV farms in both Kenya and Tanzania. These findings are consistent with similar studies conducted in Kenya and Uganda by Chepchirchir et al. [60] and in Botswana by Machekano et al. [62]. These low adoption rates could be attributed to the high associated costs and technical proficiency required in implementing some biological control methods [60]. It was also discovered that only a few of the vegetable farmers surveyed in both countries intercropped their vegetables with pest-repellant crops, which is analogous to the study conducted by Nampeera et al. [63].

Several agroecological practices were assessed in this study, such as handpicking infested vegetables, crop rotation, intercropping with pest-repellant crops, and the use of traps. Other techniques included the use of parasitoids, the use of predators, the application of biopesticides, and the use of pest-resistant cultivars. Interestingly, most of the farmers who were aware of these practices recognized that they were effective. However, the number of farmers who were aware of each of these practices was always proportionately higher than those who implemented them on their farms. This could demonstrate farmers' negative attitudes toward these agroecological practices [68]. A study by Timprasert et al. in Thailand comparably found that vegetable farmers had rejected similar practices due to difficulty in implementation as well as an over-valuation of the potential of synthetic pesticides in pest control [72]. A study by Van Hulst and Posthumus in Laikipia East, Kenya, comparably found compelling evidence indicating that farmers' attitude plays a significant role in their willingness to adopt some agricultural practices that were based on agroecological principles [73].

Finally, our findings show that there is a significant relationship between training provided to farmers and the adoption of some agroecological practices in both Kenya and Tanzania. These results corroborate the findings of Kanjanja et al., that training provided to farmers in Singida district, Tanzania, had a positively significant effect on the adoption of agroecological practices [33]. Generally, empirical evidence suggests that providing farmers with training through means such as demonstration plots or group membership has a significant impact on the adoption and diffusion of new practices [74–76].

## 5. Conclusions

The results of this study highlight limited knowledge, low adoption, and generally negative attitudes toward agroecological practices by smallholder vegetable farmers. A significant number of farmers lacked awareness about these practices and only a small minority in both countries had received any form of relevant training. Most of the farmers in both countries also barely practiced most of the agroecological production methods under survey, except for crop rotation and hand picking of pest-infested vegetables, and even among those who did, these two techniques were often not recognized as effective for improving vegetable production or managing pests and diseases. Furthermore, the study revealed that most of the farmers who were aware of agroecological-based practices recognized them as being effective, but the majority still did not implement them on their farms, indicating farmers' negative attitudes and reservations towards these practices.

Our findings underscore a need to increase farmers' awareness of the benefits of agroecological farming through targeted awareness campaigns and enhancing access to training programs on agroecological vegetable production methods and pests and disease management. We recommend increased farmer-to-farmer knowledge sharing to provide a platform for farmers to exchange their indigenous knowledge within the community. We also endorse initiatives such as farmer cooperatives and market linkages that empower smallholder farmers and increase their participation along the value chain. Furthermore, we recommend the development of policies that promote and support agroecological farming approaches such as regulations that prohibit or restrict the use of harmful chemicals and

encourage the use of organic inputs. We strongly encourage further research to complement these findings and facilitate the development of effective interventions aimed at promoting the adoption of various agroecological practices. An empirical study using experimental tools is also needed to validate the results of the vegetable intercropping approach reported by farmers.

**Supplementary Materials:** The following supporting information can be downloaded at: https://www.mdpi.com/article/10.3390/su152316491/s1: Figure S1: Types of crucifers and traditional African vegetables grown in Kenya and Tanzania; Figure S2: The years in which farmer training on vegetable agroecological production was carried out in Kenya and Tanzania; Figure S3: Organizations that trained vegetable farmers on agroecological production in Kenya and Tanzania; Table S1: Types of plants used by vegetable farmers to repel different pests in Kenya and Tanzania; Table S2: Types of plants used by vegetable farmers to attract different pests in Kenya and Tanzania.

**Author Contributions:** Conceptualization, E.C.K., M.M.K., D.M.M., D.O.O. and K.S.A.; methodology, E.C.K., M.M.K., D.M.M., D.O.O. and K.S.A.; investigation, E.C.K., M.M.K., D.M.M. and K.S.A.; data curation, E.C.K., M.M.K., D.M.M. and K.S.A.; validation, E.C.K., M.M.K., D.M.M., D.O.O. and K.S.A.; writing—original draft preparation, E.C.K. and M.M.K.; writing—review and editing, E.C.K., M.M.K., D.M.M., D.O.O. and K.S.A.; supervision, M.M.K., D.M.M., D.O.O. and K.S.A.; project administration, D.M.M. and K.S.A.; funding acquisition, D.M.M. and K.S.A. All authors have read and agreed to the published version of the manuscript.

**Funding:** This research was funded by the Biovision Foundation project "Intensified agroecological-based cropping systems to enhance food security, environmental safety, and income of small-holder producers of crucifers and traditional African vegetables in East Africa—AGROVEG" (DPP-020/2022–2024) through the International Centre of Insect Physiology and Ecology (*icipe*). The authors gratefully acknowledge the icipe core funding provided by the Swedish International Development Cooperation Agency (Sida); the Swiss Agency for Development and Cooperation (SDC); the Australian Centre for International Agricultural Research (ACIAR); the Norwegian Agency for Development Cooperation (Norad); the Federal Democratic Republic of Ethiopia; and the Government of the Republic of Kenya. The views expressed herein do not necessarily reflect the official opinion of the donors.

**Institutional Review Board Statement:** No institutional approval was required to conduct the study.

**Informed Consent Statement:** Informed consent was obtained from all subjects involved in the study.

**Data Availability Statement:** We confirm that the data and methodology employed in the research are proprietary. The derived data that support the findings of this study are available upon reasonable request from the corresponding author.

**Conflicts of Interest:** The authors declare no conflict of interest.

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
