# Peer review of "Farmers’ Knowledge, Attitude, and Practices Regarding the Use of Agroecological-Based Pest Management Practices in Crucifers and Traditional African Vegetable (TAV) Production in Kenya and Tanzania"

_sustainability, doi:10.3390/su152316491_

Round 1

Reviewer 1 Report

Comments and Suggestions for Authors Comments to Authors Comment 1: This manuscript addresses a timely topic and makes a relevant contribution to the field. However, minor revisions are needed before it can be published. Comment 2: The title is absolutely correct and clearly representing the manuscript. Comment 3: Abstract is must be modified with key findings only Comment 4: Many small grammatical errors are present in the text, which should be carefully rectified for improvement of the manuscript. General ‘English’ language improvement is recommended. Comment 5: Discussion section should be modified and should give more deep knowledge regarding the subject. Update and modify this part extensively with recent year papers and salient information. Comment 6: Modify the ‘conclusions’ part more holistically as per the updated article.

Reviewer 2 Report

Comments and Suggestions for Authors

The title of this manuscript is not suited to study results, it should be changed 

There are spelling and grammatical errors in the abstract of this paper. It cannot be sent out to a handling editor until these mistakes have been rectified.

If there are these mistakes in the abstract, what about the rest of the paper? Please pay very careful attention to the language throughout the paper

Introduction of subject experiment not given and explained. Author should explain briefly why this study is required to conduct. Results presentations not in scientific terms, they should be revised profoundly. Future directions are also not presented well overloaded text should be removed.

Keywords. These sections also should be according scientific requirement 

The overall introduction/background section is very confusing and it needs major revision as the authors have added a lot of irrelevant information. This section should be precise and to the point. Most of the literature is much old and irrelevant to the experiment which needs to be addressed. There is several overlapping information in this section, authors are advised to remove such information.

The manuscript has various typographical and grammatical errors which must be corrected. Therefore, the authors have better get English language assistance to modify this manuscript.

Reference are too old in the introduction and discussion section, In so doing, it is suggested that the following articles be used as a reference.1.Pyrosequencing uncovers a shift in bacterial communities across life stages of Octodonta nipae (Coleoptera: Chrysomelidae)." Frontiers in microbiology 10 (2019): 466. 2. Role of primary metabolites in plant defense against pathogens

Comments on the Quality of English Language

Lot of typo and grammar errors throughout the manuscript 

Reviewer 3 Report

Comments and Suggestions for Authors

The authors studied that farmers knowledge, attitude, and practices on use of agroecological-based pest management practices in crucifers and traditional African vegetables production in Kenya and Tanzania, using the structured and pretested questionnaires. They found that limited knowledge, low adoption, and generally negative attitudes hindered agroecological practices by smallholder vegetable farmers. Based on those results, authors gave some substantive suggestions.

This manuscript is very beneficial for cultivation and quality improvement of vegetable in Kenya and Tanzania. Generally, this manuscript is well organized, and I have no more comments except two suggestions.

(1) Generally speaking, when citing a literature in the text, it is necessary to list the   surname of the first author of the literature. Take an example, the sentence According to [14] the frequency of application of synthetic pesticides and insecticides by farmers in a bid to mitigate some of these constraints has been increasing every year ” in Line 52-53, is best to be revised into According to Macharia et al., the frequency of application of synthetic pesticides and insecticides by farmers in a bid to mitigate some of these constraints has been increasing every year [14], or The frequency of application of synthetic pesticides and insecticides by farmers in a bid to mitigate some of these constraints has been increasing every year [14].

This situation frequently appears in the text of manuscript, such as [50] in Line 329, [58] in Line 347, [60] in Lines 352 and 363, [61] in Line 356, [62] in Line 358, [63] in Line 366, [64] in Line 370, [33] in Line 383, etc.

 (2) The tense of the sentence in Line 329 to 330 was mistaken.

Reviewer 4 Report

Comments and Suggestions for Authors

The work is interesting. Following points has to be addressed.

1. The parameters used in the questionnaire has to be mentioned 

2. The significance of the X2 and P value in the table 1, has to be discussed and explained with respect to the problem statement and the inference derived form the value has to be clearly mentioned.

3. The significance of the t-test value in the table 2, has to be discussed and explained with respect to the problem statement and the inference derived form the value has to be clearly mentioned.

4. What is the rationale behind the consideration of countries like Kenya and Tanzania?

Comments on the Quality of English Language

Minor

Round 2

Reviewer 2 Report

Comments and Suggestions for Authors

authors address all concerns point by point